# Vibration Energy Signal Information for Measure Dynamic Preferences of Ceramic Building Materials Using Experimental Modal Analysis Methodology

**DOI:** 10.3390/ma15041452

**Published:** 2022-02-15

**Authors:** Mariusz Żółtowski, Gabriela Rutkowska, Michał Liss, Tomasz Kałaczyński, Martin Krejsa

**Affiliations:** 1Institute of Civil Engineering, Warsaw University of Life Sciences (SGGW), 02-787 Warsaw, Poland; gabriela_rutkowska@sggw.edu.pl; 2Faculty of Mechanical Engineering, Bydgoszcz University of Science and Technology, 85-796 Bydgoszcz, Poland; michal.liss@pbs.edu.pl (M.L.); kalaczynskit@pbs.edu.pl (T.K.); 3Department of Structural Mechanics, VSB-Technical University of Ostrava, 70800 Ostrava, Czech Republic; martin.krejsa@vsb.cz

**Keywords:** masonry elements, vibrations, diagnostics, modal analysis

## Abstract

Building constructions and their elements must meet requirements related to stability and strength. These are the conditions that determine the safety of the structure during both construction and operation. Safety assessment is based on diagnostic tests, aimed at checking the quality of the erected objects or locating damage to structural elements that arise during operation. This work focuses on experimental SISO modal analysis of ceramic masonry elements. An experiment was carried out on twenty samples of solid brick (good, and intentionally damaged). From this, it was possible to determine the variability of the obtained measurement results of the vibration characteristics of the masonry element, and, thus, to assess the value of the method used in the given research problem. The aim of this study was to test the effectiveness of assessing the degradation of the tested brick elements based on stabilization diagrams. The research confirmed the usefulness of experimental modal analysis in identifying damage to masonry elements, and it has been implemented in selected brickyards in Poland.

## 1. Introduction

The dynamic properties of building structures have a direct impact on the vibration level of the system, noise emissions, fatigue strength, and structural stability. In most cases, the analysis of dynamic properties encountered in practice is carried out on the basis of the analysis of the behavior of the structural model, because the study of real objects is not always possible. In most applications, simple identification methods are used, where changes in m, k, and c values (mass, stiffness, and damping) or changes in the parameters of amplitude–frequency characteristics (spectrum of the vibration process) are determined.

For complex, often non-linear systems, complex modal analysis (theoretical, experimental, or operational) is used. The modal model, which is an ordered set of natural frequencies, the corresponding damping coefficients, and the modes of natural vibrations, allows predicting the reaction of an object to any disturbance, in both the time and frequency domains.

New tools in the field of non-invasive tests concern the possibility of using vibration diagnostics methods and modal analysis methods, as well as modern vibration acquisition and processing, to assess the quality of elements and entire fragments of building structures. In practical applications, they allow for a better understanding of the behavior of individual tested elements and building materials, as well as complex structures, with optimization in the process of their design and assessment of hazardous conditions.

Modal analysis is used in diagnostics, modification, and monitoring of structures, as well as in the validation and control of analytical and numerical models; it is widely used in mechanical engineering, in the aviation industry, in electronics and electrical engineering, rail transport, and even in agriculture [1,2,3]. The analysis does not directly observe the effects of damage or wear, but only their signs, in the form of values of measured quantities such as stabilization diagrams or accompanying vibration estimators. Only an appropriate analysis (interpretation) of the measured vibration signals enables the recognition of the technical condition of a machine or structure elements, detection (localization) and determination of damage severity, and determination of their causes and measures.

The quality of the analysis depends on the reliability of the adopted structural model in the FEM environment, in the case of theoretical modal analysis or the physical model of the real object, or its part in the case of experimental modal analysis. Modal analysis can also be carried out during normal operation on a real object, in which case it is referred to as operational [4,5].

Experimental modal analysis usually consists in forcing the movement of the examined system (object) and measuring the forcing and response. The analysis procedure can be carried out by various methods When an object is excited by a modal hammer at subsequent points and the vibration sensor (transducer) is fixed at a fixed point, or when the object is excited by a modal hammer at one point and the vibration sensor (transducer) changes position, we use the SISO (single input single output) method. In cases where there is more than one vibration sensor, this is called SIMO (single input multiple outputs) [6,7].

The purpose of this work was to conduct an experimental modal analysis using the SISO method and to confirm its usefulness in identifying damage to masonry elements, based on a comparison of the FRF function and stabilization diagrams in a damaged and undamaged object.

The frequency response function (transition function) is a ratio of output signal spectrum to the input signal, as a function of frequency, it is a composite function and can be represented by the formula [6,7,8,9,10]:(1)$$FRF\left(\omega \right)=\frac{X\left(\omega \right)}{F\left(\omega \right)},$$
where $X\left(\omega \right)$-response spectrum of the system$\;x\left(t\right)$, $F\left(\omega \right)$-input signal spectrum$\;F\left(t\right)$.

On this basis, it is possible to determine the natural frequency of the system. Whereas the coherence function expressed by the formula [10,11,12]:(2)$$\gamma_{xF}^{2}\left(f\right)=\frac{{\left|G_{xF}\left(f\right)\right|}^{2}}{G_{xx}\left(f\right)G_{FF}\left(f\right)}$$
where $G_{xF}\left(f\right)$-reciprocal power spectral density between input (reference) signal $F\left(t\right)$ and exit signal (answers) $x\left(t\right)$,$G_{FF}\left(f\right)$-the spectral density of the input signal power $F\left(t\right)$,$G_{xx}\left(f\right)$-the spectral density of the response signal power $x\left(t\right)$, describes the similarity of two signals as a function of frequency. If signals $x\left(t\right)$ and $F\left(t\right)$ come from the same source, the coherence function always assumes a value of 1.

Modal analysis has been used for many years in technical mechanics; in construction it is less widespread. In science publications by many authors [11,12], it has been proven that it is possible to use this methodology in research evaluation of mechanical properties in various structures. Later, authors such as [13,14] adapted this research methodology to civil engineering. Many years of tests conducted on real building structures made it possible to create a new methodology dedicated to the use in research of different civil engineering elements, including masonry elements.

## 2. Materials and Methods

The subject of this study are ceramic masonry elements in the form of solid brick and checker brick (Figure 1). Ten intact and 10 damaged samples of each type of brick were tested. Damage was caused (forced) by a testing machine, by applying an increasing compressive force to the sample axially in the direction of the Z axis, until the first signs of damage appeared, as seen in the figures.

The research was performed on building material in the form of a solid ceramic brick of type L (veneer) B (without holes), as shown in Figure 1. Ceramic brick is made in class 20, which corresponds to a compressive strength of 20 MPa. The ceramic brick was 25 cm long, 12 cm wide, and 6.5 cm high. The ceramic brick had even sides, uniform color, and no cracks and nicks.

The modal test was carried out by forcing the samples to vibrate with a modal hammer blow. The SISO method was used, in which the strokes were made at one point and the location of the response changes [13]. The sample was suspended on an unstretched fishing line, which allowed all bonds to be released, and then the vibrations were forced from above in the direction of the Z axis; because from a practical point of view, the information regarding the transition of the vibration signal in the direction consistent with the direction of action of the compressive forces of the wall made of the tested element is important. The signal was received using a piezoelectric sensor glued onto a given wall of the sample. Response measurement points were determined and measured 10 times each, which allowed obtaining an average unloaded value of frequency characteristics [14,15].

SIEMENS LMS Test.Xpress measuring equipment (Figure 2) was used to measure the time histories of system excitation and response. This software makes it easy to carry out modal analysis of brick elements, as well as any other building structures.

As a result of the tests, time courses of excitation force (modal hammer) and time courses of response (piezoelectric sensor) were obtained, as well as their visualizations.

During the vibration tests, the possibility of acquiring new cognitive values that could be useful for assessing the destruction of the tested masonry elements was analyzed. Such information, apart from the time course of extortion, may be derived from the numerical value of the surface area of the obtained frequency functions of the signal passing through the tested element [16].

### 2.1. Experimental Modal Analysis

The modal analysis method is used for examining dynamic properties; tracking parameters changes in a wall element model, resulting in wear, damage, or failure. A modal model is created in the form of a set of natural frequencies, vibration modes, and damping coefficients for an object without damage as a pattern. During operation, the modal model is identified, and compared with the undamaged object. When correlation occurs, it shows the object is fit. In the absence of correlation, the object is in a state of unfitness, caused, for example, by aging or damage [16,17].

The sets of eigenfrequencies, the damping, and the modes of free vibrations determined in the eigenvalues problem allow structural behavior simulation under excitation, control selection, structure modification, and other factors.

The analysis of eigenfrequencies and eigenvectors is obtained from the motion equation (after omitting terms containing the damping matrix and external loads vector). The natural vibration motion equation has the following form:(3)$$M\ddot{q}+Kq=0$$

For a system with one degree of freedom, the solution is:(4)$$q\left(t\right)=\vec{q}\sin\left(\omega t+\varphi \right)$$
where $\vec{q}$
is the vector of the natural vibration amplitudes.

Substituting the above equation and the second derivative into the equation of motion, we achieve:(5)$$\left(-\omega^{2}M+K\right)\vec{q}\sin\left(\omega t-\varphi \right)=0$$

The above equation must be fulfilled for any moment t; then, the system of algebraic equations is obtained:
(6)$$\begin{array}{c} \left(K-\omega^{2}M\right)\vec{q}=0\\ \left(k_{11}-\omega^{2}m_{11}\right)q_{1}+\left(k_{12}-\omega^{2}m_{12}\right)q_{2}+\ldots +\left(k_{1n}-\omega^{2}m_{1n}\right)q_{n}=0\;\\ \left(k_{21}-\omega^{2}m_{21}\right)q_{1}+\left(k_{22}-\omega^{2}m_{22}\right)q_{2}+\ldots +\left(k_{2n}-\omega^{2}m_{2n}\right)q_{n}=0\;\\ --------------------------------\\ \left(k_{n1}-\omega^{2}m_{n1}\right)q_{1}+\left(k_{n2}-\omega^{2}m_{n2}\right)q_{2}+\ldots +\left(k_{nn}-\omega^{2}m_{nn}\right)q_{n}=0\; \end{array}$$

The result is a linear homogeneous algebraic equations system that has a non-zero solution, only when
(7)$$\det\left(K-\omega^{2}B\right)=0$$

After transformations, we obtain an nth degree polynomial with respect to$\;\omega^{2}$. Multiple roots can occur among the roots, and the vector made up of a frequency set arranged in an increasing value order is called the frequency vector, and the first frequency is called the fundamental frequency [6].
(8)$$\omega =\left[\omega_{1},\omega_{2},\ldots ,\omega_{n}\right]$$

Experimental modal analysis was used for identifying the structure modal parameters. The identification experiment consisted of object vibration forcing with the simultaneous measurement of the exciting force and response, in the form of a vibration acceleration spectrum. The modal model was obtained from the stabilization diagram. The identification experiment in EMA is shown in Figure 3.

The measurement results were processed in the software system (SIEMENS LMS), obtaining the vibration spectrum of the driving force at the system input, the spectrum of the vibration acceleration amplitude at the system output, and stabilization diagram; from which we were able to estimate the parameters of the modal model [6,8].

A TSD (traditional stabilization diagram) is a graph where the frequencies are plotted as the X-axis and the model orders as the Y-axis. It shows the poles of a system at different model orders. Physical poles occur at the same frequency at increasing model orders, forming a vertical column of poles, whereas the spurious numerical poles will not stabilize during this process and can be disregarded more easily. The modal parameters are computed from the physical modes in the same order. It should be noted that the most crucial step of using a TSD to identify parameter identification is selecting a stable vertical column of poles (Figure 4).

### 2.2. Measurement Systems

The SIEMENS LMS SCADAS Recorder system is one of the most advanced measurement systems using the modal analysis methodology in research of degradation.

It is a measurement system for data acquisition, analysis, and reporting. It includes procedures dedicated to structural and acoustic testing, environmental testing, and quality testing.

Test.Lab is used to provide data collected on real objects and integrate them into the simulation process. These data are described as time waveforms, stabilization diagrams, cross power waveforms, etc.

The condition of masonry elements should be tested with a simple and effective method, using the minimum number of measurements. This software allows easily conducting, according to the developed degradation state analysis algorithm, modal analysis of masonry elements and any other building structures.

Correct measurement depends on obtaining the level of the exciting force that was previously determined and on an appropriate level of response signal. Repeatability tests of waveforms concerning measurements of changes in material destruction state are the biggest advantages of the SIEMENS LMS measurement system.

## 3. Results and Discussion

### 3.1. Modal Analysis Results of Solid Brick Samples

The object of this research was a building material in the form of a solid ceramic brick of type L (veneer) B (without holes), shown in Figure 5. The ceramic brick was made to class 20, which corresponds to a compressive strength of 20 MPa. The ceramic brick was 25 cm long, 12 cm wide, and 6.5 cm high. The ceramic brick had even sides, uniform color, and no cracks and nicks.

### 3.2. Research Methodology

The modal study was performed on a full ceramic brick using experimental modal analysis. In this method, using a modal hammer (PCB 086C03), structural vibrations of the test object were induced, registering them with a uniaxial piezoelectric vibration sensor (PCB 333B50). The measurement points were placed on the research object in accordance with the diagram presented below—Figure 5.

Excitation of the test object with vibrations was carried out at point C1. Two waveforms were recorded in the experiment: one from the modal hammer, and the other from the vibration sensor attached to the test object. As a result of the recorded time courses, spectral functions of the FRF transition were obtained (Figure 6). At each of the measurement points, the obtained FRF waveform was the result of averaging 10 successive blows with a modal hammer. On this basis, a stabilization diagram was constructed, from which individual modal parameters were estimated in the form of natural frequency, modal damping coefficients, and vibration modes. Based on these parameters, a modal model of an undamaged ceramic brick was created; thus, completing the first stage of modal research of the object. The next, second, stage of modal tests consisted in subjecting a full ceramic brick to a compression strength test on an Instron testing machine. A load of 30 MPa was applied on the testing machine. This value is 10 MPa higher than the maximum compressive strength of a full brick. The endurance test was carried out until the first symptoms of damage to the test object were noticed in the form of cracks, while not leading to the complete destruction of the brick. Figure 7 shows the cracks of the solid brick obtained after the strength test. The full ceramic brick with the fracture was then subjected to a modal test, to observe changes between the individual modal models.

The crack in the solid ceramic brick was between measuring points C1 and C2, and C7 and C8.

In the modal tests, a frequency range from 0.7 Hz to 6375 Hz was adopted, due to the theoretical first natural frequency for a full ceramic brick occurring at approx. 2927.63 Hz (and 2927.68 Hz based on the tables).

## 4. Results of the Modal Study

### 4.1. Undamaged Solid Ceramic Brick

Measurement results for an undamaged solid brick were obtained from a total of eight measuring points, four on each side of the brick. Figure 8 shows the average FRF plot from the eight measurement points, on which the calculated theoretical first frequency of natural vibrations was based. There was only one peak (2875.97 Hz) in the vicinity of this value, which could indicate convergence of these frequencies. Based on this diagram, it can also be observed that at this frequency there is a clear change of the vibration phase, which also indicates the presence of structural vibrations of the test object at this place.

Based on the measured characteristics of the FRF, a stabilization diagram was created (Figure 9) for an undamaged solid ceramic brick. The size of the modal model was limited to 128, assuming the multivariant option. Figure 10 shows a stabilization diagram with numerous stable poles that may constitute potential structural vibrations of the research object.

To confirm the selected stable poles, which presumably may be structural vibrations of a solid brick, validation of the selected poles and the corresponding forms of free vibration should be performed. The validation was carried out using the AutoMAC function, the results of which, in relation to the analyzed object, are shown in Figure 11. On this basis, five characteristic modes of free vibrations of the undamaged full ceramic brick were selected, and are summarized in Table 1. List of natural frequencies and damping coefficients for undamaged solid ceramic brick are shown in Table 2.

Each of the determined frequencies of natural vibrations has an appropriate form of these vibrations. The individual modes of the free vibrations for the undamaged full ceramic brick in their maximum deflection from the equilibrium position are presented in the figure below.

### 4.2. Damaged Full Ceramic Brick

Measurement results for a damaged full brick were obtained from a total of eight measuring points, four on each side of the brick. Figure 14 shows the average FRF plot from the eight measurement points, on which the calculated theoretical first frequency of natural vibrations was based. There was only one peak (3053.40 Hz) in the vicinity of this value, which would indicate a convergence of these frequencies. Based on this diagram, it can also be observed that at this frequency there is a clear change of the vibration phase, which also indicates the presence of structural vibrations of the test object in this place.

Theoretical first natural frequency of brick vibrations are verry good shown in FRF function diagram—Figure 12.

Based on the measured characteristics of the FRF, a stabilization diagram was created (Figure 13) for the damaged full ceramic brick. The size of the modal model was limited to 128, assuming the Multivariant MIF option. Figure 15 shows a stabilization diagram with numerous stable poles that may constitute potential structural vibrations of the research object.

To confirm the selected stable poles, which presumably may be structural vibrations of a solid brick, validation of the selected poles and the corresponding forms of free vibration should be performed. The validation was carried out using the AutoMAC function, the results of which, in relation to the analyzed object, are shown in Figure 14. On this basis, six characteristic modes of free vibration of the damaged full ceramic brick were selected, which are summarized in Table 3. List of natural frequencies and damping coefficients are shown in Table 4.

Each of the determined frequencies of natural vibration has an appropriate form of vibration. The individual modes of free vibrations for a damaged full ceramic brick in their maximum deflection from the equilibrium position are presented in the Figure 15 below.

## 5. Discussion

A comparison of the generated diagrams surely shows that it was possible to identify damage to the tested elements. This is evidenced by the diversity in the very repetitive graphic appearance of the tested vibration functions, depending on the suitability or damage of the tested element. There are several examples of a noticeable difference in the numerical values of the surface areas, depending on the damage of the tested masonry element. This proves that by measuring the FRF function at various time points in the object (structure) life, it is possible to detect the occurrence of damage. The modal analysis methodology is mostly used in the study of global structure, or a specific body constructed, for example, of ceramic bricks. The purpose of this article is also to show that by using modal analysis in local studies of individual brick elements, it is possible to obtain reliable information about local damage.

## 6. Conclusions

The presented research results indicate that modal analysis can distinguish between material properties, which affects the ability to distinguish between their strength properties or degradation state. The research confirmed the usefulness of experimental modal analysis in identifying damage to masonry elements, based on a comparison of the course of the FRF function and stabilization diagrams in a damaged and undamaged object. The usefulness of vibration analysis in the study of degradation (quality) of building structures, as well as their elements, results from the fact that vibration processes reflect the physical phenomena occurring in structures and their elements, on which the proper functioning or the degree of destruction depends. Practically, this type of vibration testing of single bricks using the methodology of modal analysis has been implemented in selected brickyards in Poland. It accompanies destructive tests for assessing the quality of building materials fired in these plants.

## Figures and Tables

**Figure 1 materials-15-01452-f001:**
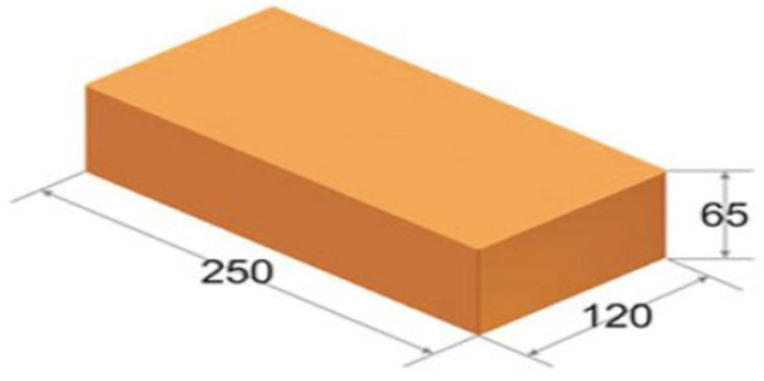
View of solid brick samples subjected to measurements.

**Figure 2 materials-15-01452-f002:**
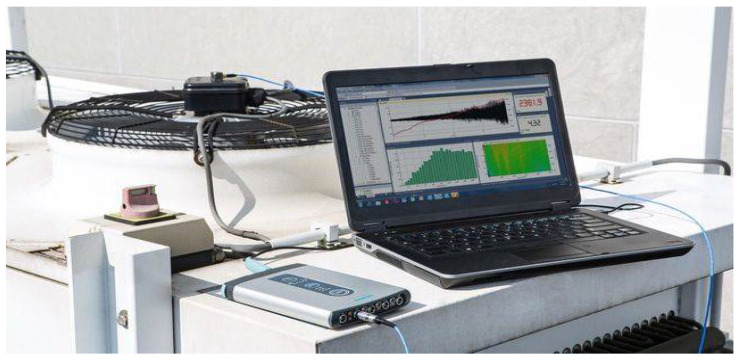
SIEMENS LMS Test.Xpress measuring equipment.

**Figure 3 materials-15-01452-f003:**
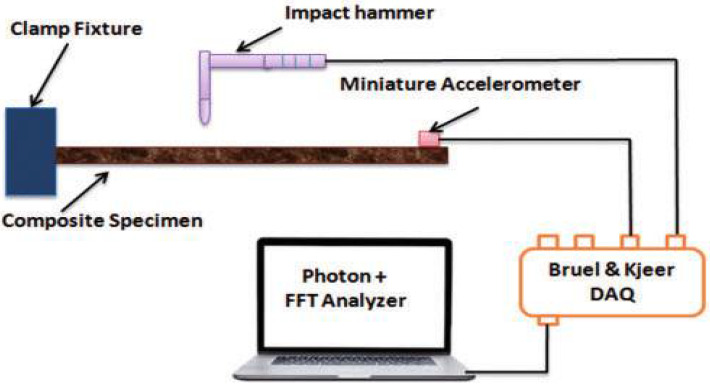
The essence of research in experimental and operational modal analysis.

**Figure 4 materials-15-01452-f004:**
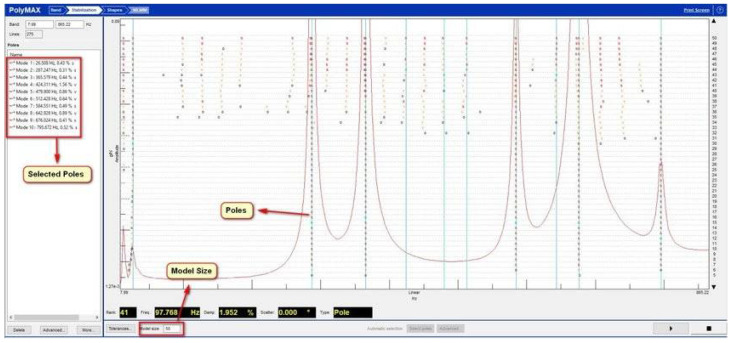
Stabilization diagram. Notations: order = order of the pole, o = unstable pole, f = pole has a constant frequency. v = pole has a constant frequency and modal vector; s—stable pole.

**Figure 5 materials-15-01452-f005:**
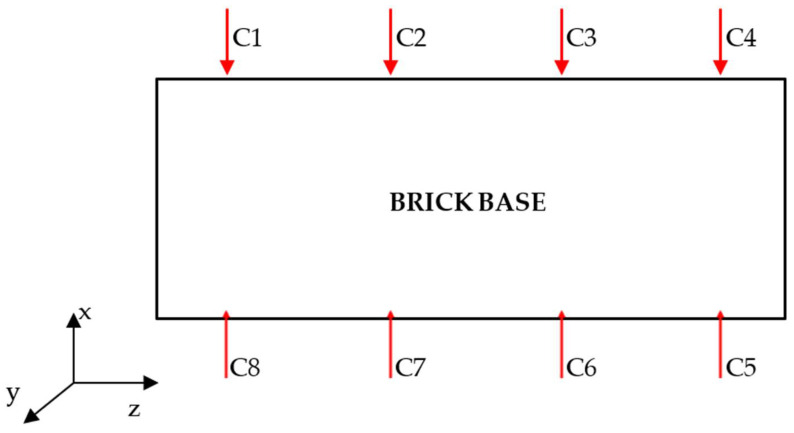
Distribution of measurement points on the research object.

**Figure 6 materials-15-01452-f006:**
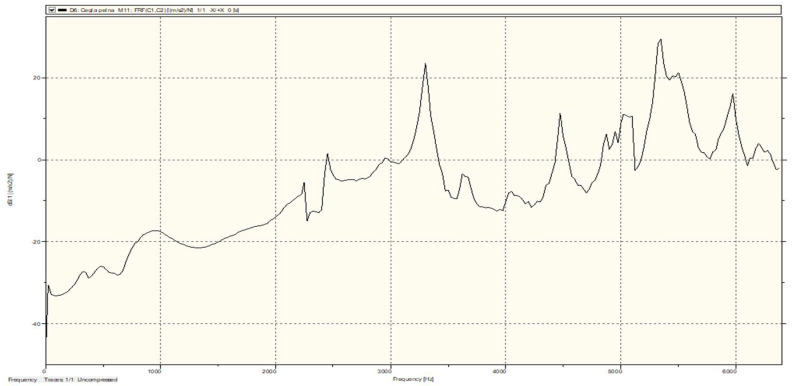
Spectral function of the FRF transition registered at point C1.

**Figure 7 materials-15-01452-f007:**
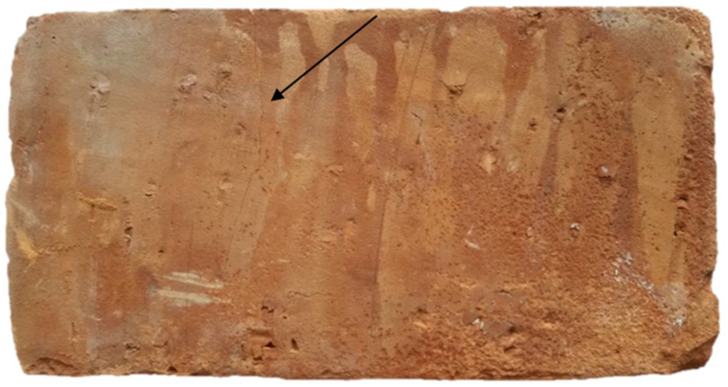
Clear fracture of the full ceramic brick after the strength test.

**Figure 8 materials-15-01452-f008:**
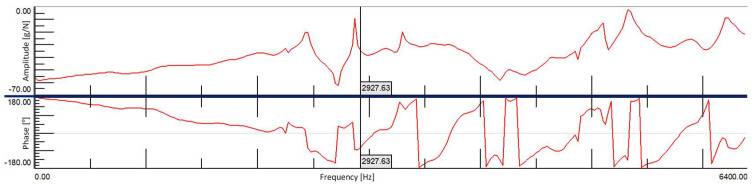
The theoretical first natural frequency of brick vibrations in the averaged FRF plot.

**Figure 9 materials-15-01452-f009:**
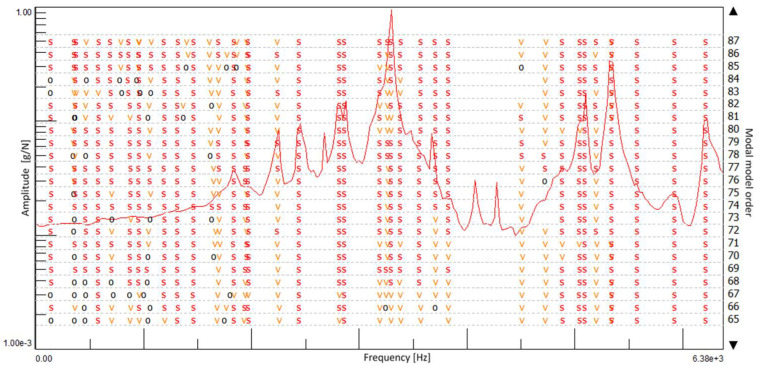
Stabilization diagram made with the use of the SUM indicator with eight measurement points for an undamaged full ceramic brick.

**Figure 10 materials-15-01452-f010:**
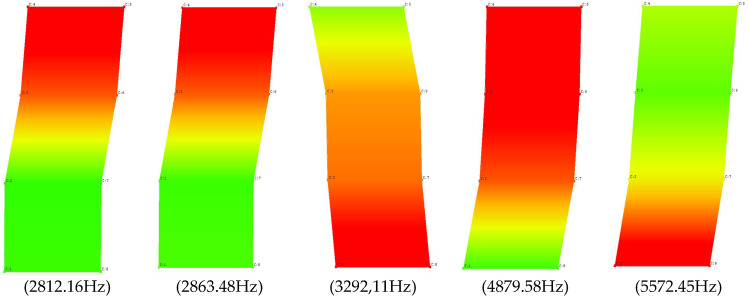
Natural frequency values of the undamaged solid ceramic brick.

**Figure 11 materials-15-01452-f011:**
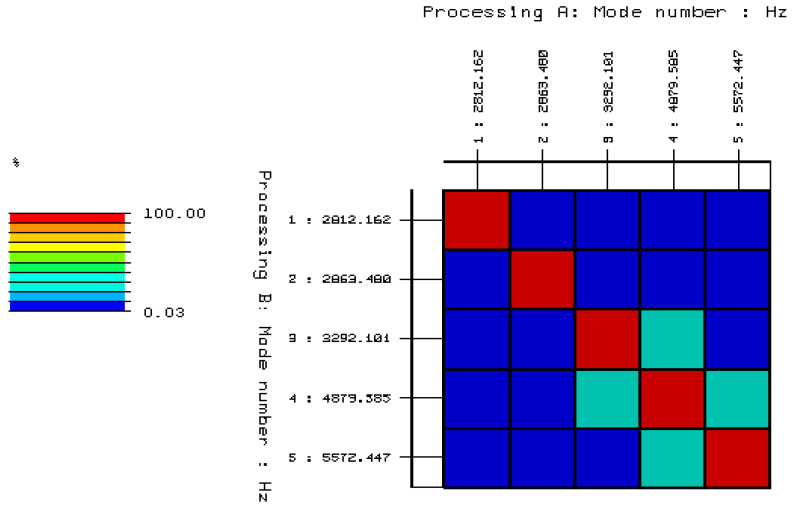
Validation of selected stable poles and the corresponding modes of free vibrations using the AutoMAC function.

**Figure 12 materials-15-01452-f012:**
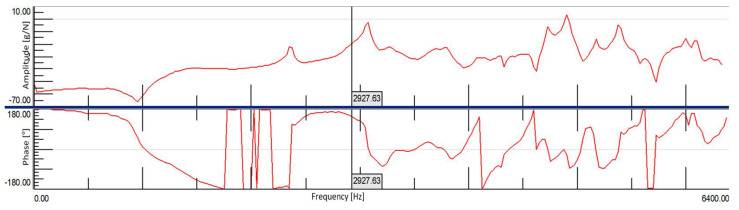
Theoretical first natural frequency of brick vibrations on the averaged FRF plot.

**Figure 13 materials-15-01452-f013:**
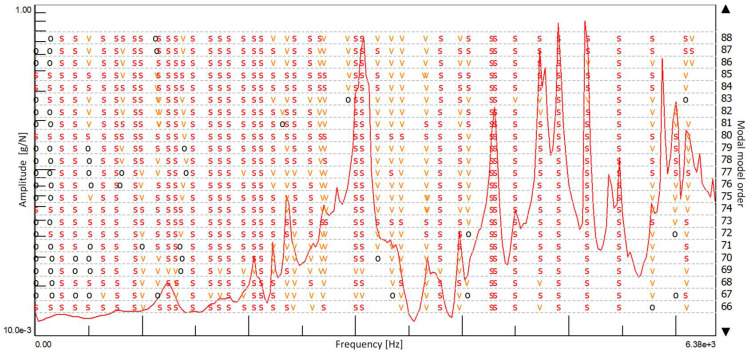
Stabilization diagram created with the use of the SUM indicator from eight measurement points for a damaged full brick because of the compressive strength test.

**Figure 14 materials-15-01452-f014:**
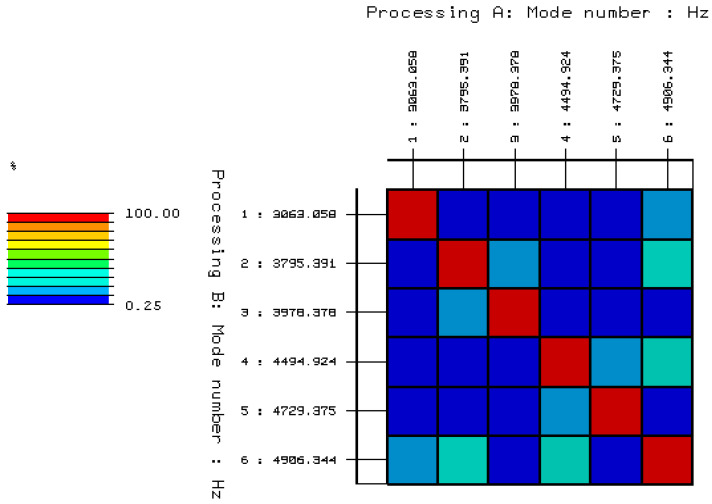
Validation of the selected stable poles and the corresponding modes of free vibration using the AutoMAC function.

**Figure 15 materials-15-01452-f015:**
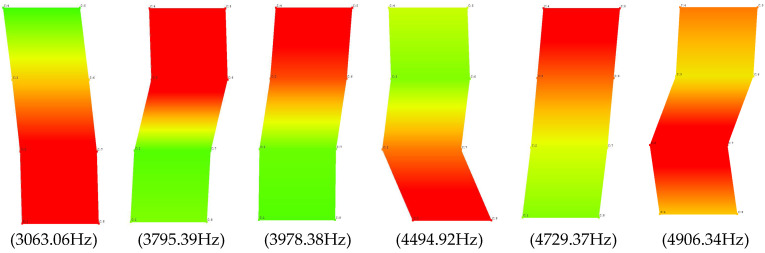
Natural frequency values of the damaged solid ceramic brick.

**Table 1 materials-15-01452-t001:** The result of the validation (AutoMAC) in tabular form for undamaged solid ceramic brick.

Undamaged Full Brick		MODE 1	MODE 2	MODE 3	MODE 4	MODE 5
		2812.16	2863.48	3292.11	4879.58	5572.45
MODE 1	2812.16	100	1.49	6.11	0.03	7.77
MODE 2	2863.48	1.49	100	2.92	7.25	1.91
MODE 3	3292.11	6.11	2.92	100	25.97	0.20
MODE 4	4879.58	0.03	7.25	25.96	100	24.29
MODE 5	5572.45	7.77	1.91	0.20	24.28	100

**Table 2 materials-15-01452-t002:** List of natural frequencies and damping coefficients for undamaged solid ceramic brick.

	Undamaged Solid Brick	
Natural Vibrations	Frequency	Damping
	(Hz)	(%)
**Mode 1**	2812.162	0.44
**Mode 2**	2863.480	0.82
**Mode 3**	3292.101	0.13
**Mode 4**	4879.585	0.14
**Mode 5**	5572.447	0.34

**Table 3 materials-15-01452-t003:** The result of the validation (AutoMAC) in tabular form for the damaged full ceramic brick.

Damaged Full Brick		MODE 1	MODE 2	MODE 3	MODE 4	MODE 5	MODE 6
		306.06	3795.39	3978.38	4494.92	4729.37	4906.34
MODE 1	2812.16	100	6.10	0.25	0.45	0.37	16.17
MODE 2	2863.48	6.10	100	18.58	0.76	0.51	31.12
MODE 3	3292.11	0.25	18.59	100	1.67	0.36	3.34
MODE 4	4879.58	0.45	0.76	1.67	100	12.80	22.03
MODE 5	5572.45	0.37	0.51	0.36	12.80	100	2.46
MODE 6	4906.34	16.17	31.12	3.34	22.03	2.46	100

**Table 4 materials-15-01452-t004:** List of natural frequencies and damping coefficients for a damaged full ceramic brick.

	Damaged Solid Brick	
Natural Vibrations	Frequency	Damping
	(Hz)	(%)
**Mode 1**	3063.058	0.23
**Mode 2**	3795.391	0.72
**Mode 3**	3978.378	0.13
**Mode 4**	4494.924	0.29
**Mode 5**	4729.375	0.12
**Mode 6**	4906.344	0.16

## Data Availability

Data are not publicly available. The data may be made available upon request from the corresponding author.

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
