# Peer review of "Vibration Energy Signal Information for Measure Dynamic Preferences of Ceramic Building Materials Using Experimental Modal Analysis Methodology"

_materials, 2022, doi:10.3390/ma15041452_

Round 1
Reviewer 1 Report
This manuscript reports an interesting study of using experimental modal analysis to characterize the structural condition of ceramic masonry bricks. 20 samples with pristine and damaged conditions were tested. Impulse vibrational responses were collected using impact hammer and distributed accelerometers. A standard modal analysis was performed. It is unclear the contribution/novelty of this manuscript. It is recommended to consider submitting to another journal for publication.
Author Response
Dear Reviewer,
Thank You VERY MUCH for conducting the review.
We tried to take into account all the comments contained in the review.
We kindly ask for Your understanding and favor, for us it is VERRY important to Us topublish this article.
Best regards
Authors
Reviewer 2 Report
The physical background of the problem considered is clear. The manuscript has a good proposal. However, I recommend the manuscript should be addressed in the revised version, are reported as follows:
1) The motivation of this work should be further stated;
2) The physical problems' background should be clarified;
3) The readability of the paper needs to be improved;
4) For general readers, authors are encouraged to discuss other kinds of materials such as: [(a) “Influence of the visco-Pasternak foundation parameters on the buckling behavior of a sandwich functional graded ceramic–metal plate in a hygrothermal environment”; (b)“An original four-variable quasi-3D shear deformation theory for the static and free vibration analysis of new type of sandwich plates with both FG face sheets and FGM hard core”;
5) The practical applications of this present work must be presented by authors.
6) Figs. 14 and 15 should be more discussed.
7) Author should add some physical explanation to improve the quality of the paper. Conclusion section must be extended in a few words via main finding and advantages of the methodology.
Author Response
Dear Reviewer,
Thank You VERY MUCH for conducting the review.
We tried to take into account all the comments contained in the review.
We kindly ask for Your understanding and favor, for us it is VERRY important to Us to publish this article.
Best regards
Authors
Reviewer 3 Report
The manuscript entitled “Vibration Energy Signal Information for Measure Dynamic Preferences of Ceramic Building Materials using Experimental Modal Analysis Methodology” requires major revision before publication. The author mainly described experimental modal analysis for a single solid brick that FRF function and stabilization diagrams in a damaged and undamaged object was studied with single input single output method. The main fault is the modal analysis results were observed by a solid ceramic brick. However, several suggestions were made to authors to enhance the content of the manuscript. Please find the comments below.
(1) In Figure 1, I suggest that the author present the solid ceramic brick just in one time only. Remove the left one and make the optical image corresponds well the drawing.
(2) Figure 2, Figure 3, Figure 5, and Figure 6 seems unnecessary if the author wants to show extra information in the software of SIEMENS LMS Test.Xpress. By the way, the image quality is too poor to read.
(3) At line 61-64, the sentence “where ………………………. describes the similarity of two signals as a function of frequency.” is confusing. What parameter describes the similarity of two signals as a function of frequency?
(4) At line 94-95, the sentence “Response measurement points were determined and 3 were measured in each of them several times (10 times),” is confusing.
(5) At line 176, the sentence “------------ Figure 8.” is confusing with respect to the meaning of the paragraph.
(6) The weakness of the manuscript is that the single solid ceramic brick was discussed with the modal analysis. A global structure or specific body constructed by ceramic bricks was suggested to run modal analysis.
Author Response

(The authors gave the same response as above.)

Round 2
Reviewer 1 Report
The authors have done what they can to improve the quality of the manuscript. However, there is still concern about innovation. Modal analysis has been widely used in many industries over the past 3 to 4 decades. It seems the only originality is that the authors used a commercial DAQ and analysis system on a brick.
Reviewer 3 Report
After carefully read the revised version of the manuscript, I found one previous comment hasn’t been considered. In the line 196 of revised version, I still don’t get the explanation of Figure 8 when it meets line 248.